# Pancreatic cancer symptom trajectories from Danish registry data and free text in electronic health records

**Jessica Xin Hjaltelin**[1†], **Sif Ingibergsdóttir Novitski**[1†], **Isabella Friis Jørgensen**[1], **Troels Siggaard**[1], **Siri Amalie Vulpius**[1], **David Westergaard**[1], **Julia Sidenius Johansen**[2], **Inna M Chen**[2], **Lars Juhl Jensen**[1], **Søren Brunak**[1,3]*

[1]Novo Nordisk Foundation Center for Protein Research, Faculty of Health and Medical Sciences, University of Copenhagen, Copenhagen, Denmark; [2]Department of Oncology, Copenhagen University Hospital - Herlev and Gentofte, Herlev, Denmark; [3]Copenhagen University Hospital, Rigshospitalet, Blegdamsvej, Copenhagen, Denmark

**\*For correspondence:**
soren.brunak@cpr.ku.dk

[†]joint first authors

**Abstract** Pancreatic cancer is one of the deadliest cancer types with poor treatment options. Better detection of early symptoms and relevant disease correlations could improve pancreatic cancer prognosis. In this retrospective study, we used symptom and disease codes (ICD-10) from the Danish National Patient Registry (NPR) encompassing 6.9 million patients from 1994 to 2018,, of whom 23,592 were diagnosed with pancreatic cancer. The Danish cancer registry included 18,523 of these patients. To complement and compare the registry diagnosis codes with deeper clinical data, we used a text mining approach to extract symptoms from free text clinical notes in electronic health records (3078 pancreatic cancer patients and 30,780 controls). We used both data sources to generate and compare symptom disease trajectories to uncover temporal patterns of symptoms prior to pancreatic cancer diagnosis for the same patients. We show that the text mining of the clinical notes was able to complement the registry-based symptoms by capturing more symptoms prior to pancreatic cancer diagnosis. For example, 'Blood pressure reading without diagnosis', 'Abnormalities of heartbeat', and 'Intestinal obstruction' were not found for the registry-based analysis. Chaining symptoms together in trajectories identified two groups of patients with lower median survival (<90 days) following the trajectories 'Cough→Jaundice→Intestinal obstruction' and 'Pain→Jaundice→Abnormal results of function studies'. These results provide a comprehensive comparison of the two types of pancreatic cancer symptom trajectories, which in combination can leverage the full potential of the health data and ultimately provide a fuller picture for detection of early risk factors for pancreatic cancer.

## Editor's evaluation

This article presents valuable findings on the symptoms and disease trajectories preceding a diagnosis of pancreatic cancer in Denmark. The evidence is convincing and the study employed appropriate and validated methodology in line with current state-of-the-art. The work will be of interest to public health researchers and clinicians working on pancreatic cancer.

## Introduction

Pancreatic cancer has been predicted to become the second leading cause of cancer deaths, surpassing breast, colorectal, and prostate cancer (*Rahib et al., 2021*). It has few and generic symptoms resulting

**eLife digest** Pancreatic cancer is one of the deadliest cancer types. Scientists predict it will become the second largest cause of cancer-related deaths in 2030. It has few or no symptoms at early stages and often goes undetected for an extended period. As a result, patients are often diagnosed at an advanced stage when they have few treatment options and lower survival rates. Only 11 percent of patients with pancreatic cancer survive five years past their diagnosis. Earlier detection and surgery to remove the tumor increase patient survival to 42% at five years. Those who undergo surgery at the earliest stage have an 84% survival rate at five years.

Developing ways to screen for and detect pancreatic cancer early could improve patient survival. Identifying early symptoms is critical. So far, studies show links between weight loss, abdominal pain, lower back pain, and new-onset diabetes and pancreatic cancer. But clinicians often overlook these symptoms or do not associate them with cancer. National health registries may be data sources that scientists can use to zoom in on early pancreatic symptoms and create alerts for clinicians.

Hjaltelin, Novitski et al. identified potential pancreatic cancer symptoms using patient registry data and electronic health records. Hjaltelin, Novitski et al. extracted potential pancreatic cancer-related disease or symptom trajectories from 7 million patients listed in the Danish National Patient Registry. They also scoured clinical notes in 34,000 patients' electronic health records for symptoms. The electronic health records yielded more promising symptoms than the registry. But both data sources produced complementary information. The analysis showed that some symptoms, like jaundice, were associated with higher survival rates because they may lead to earlier diagnosis.

The data so far suggest that symptoms leading up to a pancreatic cancer diagnosis may be nonspecific and not occur in a particular order. As the cancer progresses, symptoms may become more specific and severe. Further assessment of the study's results is necessary. Tools like artificial intelligence or advanced text mining may allow scientists identify more definitive early symptom trajectories and help clinicians identify patients earlier.

in late diagnosis (*Chari et al., 2015*; *Kim and Ahuja, 2015*) and poor prognosis with a 5-year survival rate of 11% (*American Cancer Society, 2020*). Hence, improved knowledge of symptoms and diseases occurring early is of high importance to treat this cancer type at a curable stage and provide better prognosis and guide screening programs for pancreatic cancer (*Risch et al., 2015*). If the cancer is detected at an early stage, where surgical removal of the tumour is possible, the survival rate increases to 42% ('American Cancer Society', 2020). The patients with stage I disease have the highest 5-year survival of 83.7% (*Blackford et al., 2020*).

Symptoms of pancreatic cancer are often mistaken for signs of less severe illnesses and overlooked in clinical practice. Some of the most frequent symptoms linked to pancreatic cancer are weight loss, abdominal pain, and anorexia (*Hidalgo, 2010*; *Mizrahi et al., 2020*; *Park et al., 2021*). Others include upper abdominal pain, cholestasis, nausea (*Hidalgo, 2010*), and dark urine and thirst (*Liao et al., 2021*). New-onset diabetes has additionally been found to co-occur with pancreatic cancer when accompanied by weight loss (*Yuan et al., 2020*; *Hart et al., 2011*; *Bruenderman and Martin, 2015*).

National or regional disease registries hold longitudinal data on disease development. The registries in the Nordic countries are of high quality and among the oldest covering treatment in one-payer health care systems (*Laugesen et al., 2021*). The National Danish Patient Registry (NPR) contains hospital diagnoses since 1977 and allows for large data-driven studies to detect temporal disease progression patterns relevant in the context of stratified medicine (*Jensen et al., 2014*; *Siggaard et al., 2020*, *Lademann et al., 2019*). Recent examples are the characterization of multimorbidity correlations across cancer types (*Hu et al., 2019*) and the detection of pancreatic cancer using artificial intelligence (*Placido et al., 2023*). However, much of the deeper phenotypic patient information resides within the free text of the electronic health records (EHRs; *Jensen et al., 2012*; *Eriksson et al., 2014*; *Soguero-Ruiz et al., 2016*; *Delespierre et al., 2017*). A small-scale study using 4080 mixed types of cancers attempted to build more general 'event trajectories' using text mining and pooled analysis (*Jensen et al., 2017*). A prospective study investigating initial symptoms and diagnostic interval (time from onset to diagnosis) for known pancreatic cancer symptoms found no difference between pancreatic cancer patients and patients suspected of having pancreatic cancer (*Walter*

**Table 1.** Data set and patient characteristics.

| General cohort information | The National Patient Registry (NPR) | Electronic Health Records (EHRs) |
|---|---|---|
| Data set timeline | 1994–2018 | 2006–2016 |
| N pancreatic cancer patients | 23,592 | 3078 |
| N controls | 6.9 million | 30,780 |
| Pancreatic cancer cohort information | | |
| Female | 9328 (50.4%) | 1506 (48.9%) |
| Male | 9195 (49.6%) | 1572 (51.1%) |
| Mean age at diagnosis (female/male) | 73/69 | 72/70 |
| Age distributions (years) | | |
| <40 | 139 (0.8%) | 16 (0.52%) |
| 40–50 | 733 (4.0%) | 107 (3.48%) |
| 50–60 | 2523 (13.6%) | 352 (11.4%) |
| 60–70 | 5288 (28.5%) | 941 (30.6%) |
| 70–80 | 6017 (32.5%) | 1037 (33.7%) |
| >80 | 3821 (20.6%) | 625 (20.3%) |

*et al., 2016*). It has also been suggested that symptoms appear sporadically, adding to the complex nature of the disease manifestation (*Evans et al., 2014*). Other studies used primary care EHRs to detect pancreatic cancer symptoms and found jaundice (*Stapley et al., 2013*), back pain, lethargy, and new-onset diabetes to be linked to pancreatic cancer (*Keane et al., 2014*). These studies detected pancreatic cancer symptoms using a single-disease approach, not considering the temporal ordering of symptoms or diseases.

In this paper, we present a large-scale study to investigate pancreatic cancer symptoms longitudinally. We cover all symptoms included in the International Classification of Disease (ICD-10) terminology symptom chapter 18. Additionally, we also include in the text mining vocabulary other known or suggested pancreatic cancer symptoms. We generate and compare disease and symptom trajectories using registry data and clinical notes in EHRs to characterize the temporal ordering of symptoms across data sources.

## Results

### Extracting patient-level data from the Danish National Patient Registry and free text electronic health records

The Danish National Patient Registry (NPR) data spans the period 1994–2018, while the electronic health records (EHRs) used here are from 2006 to 2016. The NPR includes 6.9 million patients where 23,592 patients are diagnosed with pancreatic cancer. Of these we used 18,523 patients which also were confirmed by the Danish Cancer Registry. (*Table 1*). For a subset of 3078 patients, we extracted their corresponding EHRs. Almost as many females as males are identified with pancreatic cancer both in NPR and the EHRs (*Table 1*).

We text-mined the pancreatic cancer patient symptom history in clinical notes five years prior to the cancer. We compared these symptoms to the most frequent symptoms in the NRP (*Figure 1*). These are counts and show an overview of identified symptoms, which could also be present in the control population. The text mining approach was able to identify 16 unique symptoms in the clinical notes, not registered in NPR (*Figure 1A*). The most frequent symptoms exclusively found in the clinical notes were conditions related to 'Intestinal disorders', 'Dorsalgia' and 'Embolisms' (*Figure 1B*). NPR contained 24 ICD-10 symptoms not found by text mining (*Figure 1A*). Frequent symptoms exclusively identified in the NPR data comprised findings related to 'Functional capacity',

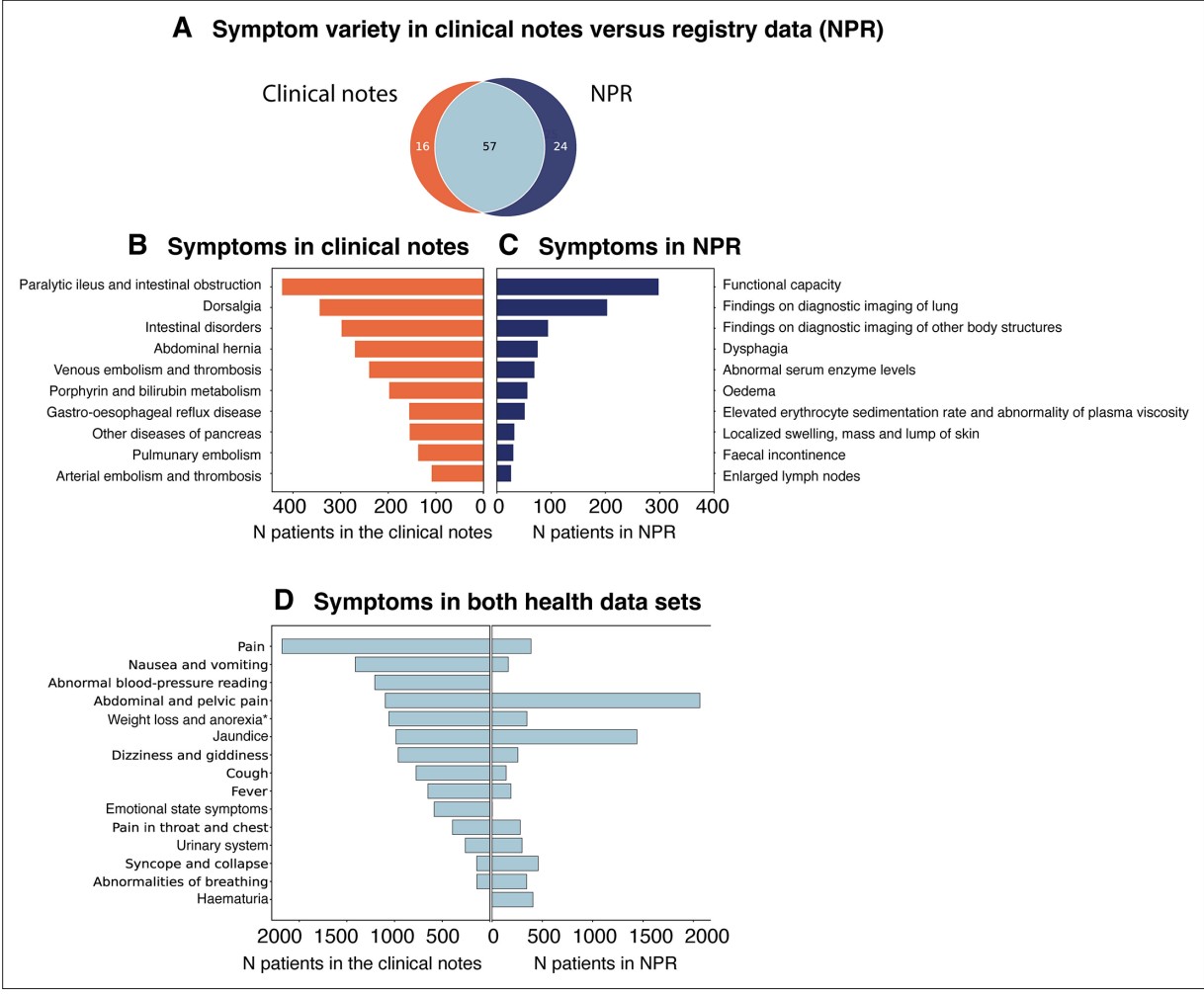

**Figure 1.** Comparison of pancreatic cancer symptoms in the Danish National Patient Registry (NPR) and electronic health records (EHRs). (**A**) Symptoms before the pancreatic cancer diagnosis identified in NPR ($N_{NPR}$ = 24), by text mining of clinical notes from EHRs ($N_{notes}$ = 16) or in both data sources ($N_{both}$ = 57). (**B**) The top 10 most frequent symptoms that are only found in the clinical notes. (**C**) The top 10 most frequent symptoms in NPR. (**D**) The most frequent symptoms from the clinical notes and NPR, from the list of 57 overlapping symptoms. *Some symptom names have been shortened for overview (see **Supplementary file 1b**).

The online version of this article includes the following figure supplement(s) for figure 1:

**Figure supplement 1.** Comparing symptoms found from text mining, registry (NPR) and established well-known symptoms of pancreatic cancer.

'Dysphagia', 'Abnormal laboratory findings', 'Oedema', and 'Abnormal findings on medical images' (**Figure 1C**). In NPR and free text clinical notes, 57 symptoms were identified that occurred in both sources. Of these, the 15 most frequent symptoms from the free text clinical notes and the NPR were compared (**Figure 1D**). Amongst these, we could identify 'Pain', 'Nausea and vomiting', 'Abnormal blood-pressure reading', 'Abdominal and pelvic pain', and 'Symptoms and signs concerning food and fluid intake'. From the hierarchical structure of the ICD-10 chapters, different levels of coding detail can be retrieved. In the symptom block 'Symptoms and signs concerning food and fluid intake', the majority of patients represent the subgroups 'Abnormal weight loss' (R63.4) and 'Anorexia' (R63.0). 'Nausea and vomiting', 'Abnormal blood-pressure reading' and 'Weight loss' were frequent symptoms in the free-text clinical notes, but have low occurrences in NPR. On the contrary, 'Abdominal and pelvic pain' was more frequent in NPR than in the clinical notes (**Figure 1D**). We also compared these symptoms to well-known pancreatic cancer symptoms (**Figure 1—figure supplement 1**) showing that most of these are also identified via the two data types.

## Temporality of symptoms from clinical notes

The distribution of symptoms registered over time can be seen for the most frequent symptoms that occur significantly more often in the pancreatic cancer patients compared to the matched control cohort (*Figure 2*). The symptom distributions are covering a five-year period before the pancreatic cancer diagnosis; the number of patients and p-values included in these can be found in *Supplementary file 1c*. If a symptom is registered multiple times in one hospital encounter it is only included once. All occurrences of a symptom during the five-year period are included for a patient. Some of the most frequent symptoms identified are symptoms related to 'Pain', 'Abdominal and pelvic pain', 'Nausea and vomiting', and 'Jaundice'. Additionally, we also found 'Fever', 'Intestinal obstruction' and 'Anorexia' to be frequent and significant symptoms among the pancreatic cancer patients.

Symptoms such as 'Jaundice', 'Disorders of porphyrin and bilirubin metabolism', and 'Intestinal obstruction' were observed closest to the pancreatic cancer diagnosis. The more general symptoms such as 'Pain', 'Intestinal disorders', 'Heartburn', and 'Fever' were found more distant in time to the pancreatic cancer diagnosis. 'Hepatomegali and splenomegali' was found to have the most differing distributions in time between the short and longer survival patient groups. For the short survival death group (≤90 days), this symptom was detected closer to the cancer diagnosis, whereas the longer survival group had the symptom detected much further back in time to the cancer diagnosis. A similar pattern is seen for 'Intestinal disorders' and 'Heartburn'.

We added cancer staging information from the Danish cancer registry (*Gjerstorff, 2011*) and calculated the stage via the pancreatic cancer TNM version 7 classification, showing that many of the patients have unknown staging information. However, we do see that there is a slight over representation of longer survival patients in the Jaundice symptoms patient group.

## Text mining validation

In total, 2867 out of 3078 (93%) of the pancreatic cancer patients in clinical notes have at least one symptom identified by text mining. A control group of 30,780 patients was generated by matching birth year, diagnosis year, age and sex. From these, 25,085 patients (82%) had a match for at least one symptom from a general symptom dictionary generated by combining general symptoms from the (1) ICD-10 chapter 18, (2) known pancreatic cancer symptoms from the literature, and (3) a systematic review of clinical notes for pancreatic cancer patients. Text mining the clinical notes identified 73 unique level-3 coded symptoms for the pancreatic cancer cases and 79 unique symptoms for the control group. The performance of the text mining methodology, using symptoms in clinical notes, was validated to check that the text mining method can correctly identify general symptoms. We used a test corpus comprising a random extraction of 200 clinical notes from 200 different patients. In these notes, a total of 807 symptoms were manually annotated and the text mining method was able to identify 675 correctly and 132 symptoms were not found. This yielded a sensitivity score of 83.4%. A specificity score of 99.8% was obtained since most clinical notes comprise non-symptom words describing the patient's contact with the hospital. Symptoms incorrectly matched to the dictionary (false positives) constitute a total of 53 words and for example captured symptoms where the text mining did not catch the meaning of a negation.

## Pancreatic cancer trajectories from NPR

Longitudinal disease trajectories were generated for the pancreatic cancer patients, where significant directional diagnosis pairs were joined to represent patients that traverse a complete disease path. The width of the trajectories illustrates the size of a patient group that moves through a particular path. A patient can follow multiple paths. The ICD-10 disease codes from NPR were used to generate disease-wide trajectories (*Figure 3—figure supplement 1*). In total 3863 patients followed at least one symptom trajectory which corresponds to 21% of all the pancreatic cancer cases (*Supplementary file 1d*). For comparison purposes, we show only significant symptom trajectories with codes from chapter 18 'Symptoms, signs, and abnormal clinical and laboratory findings' in *Figure 3*. Most of the codes from the symptoms chapter appear after the diagnosis of pancreatic cancer, such as 'General disability', 'Malaise and fatigue', and 'Cachexia', whereas fewer symptoms are found prior to the cancer such as 'Jaundice', 'Haemorrhage', and 'Pain' (throat, chest, abdomen, and pelvis; *Figure 3A*). We picked 90 days to distinguish between the early and late death patient groups to investigate

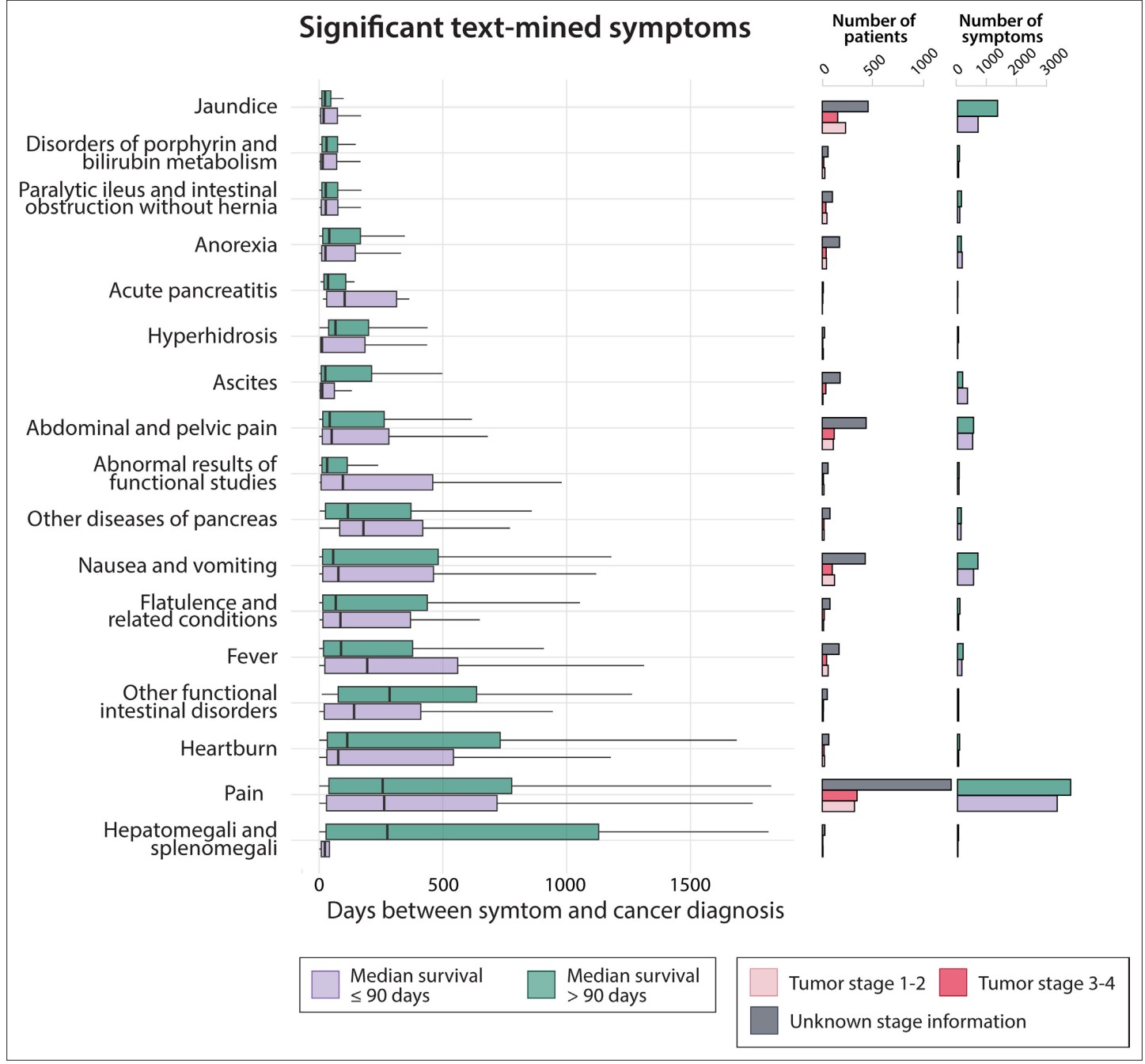

**Figure 2.** The most frequent text-mined symptoms from the clinical notes 5 years prior to pancreatic cancer diagnosis. The most common and significant (p<0.05) symptoms in the text-mined clinical notes are shown with survival information and time to pancreatic cancer diagnosis (Supplementary file 3). The symptoms are extracted over a 5-year period up to the time of the first pancreatic cancer diagnosis. If a symptom is noted more than once in one hospital encounter, the symptom is counted once only. The purple bars indicate patients with survival ≤90 days and the green bars indicate patients with a survival >90 days. Symptom names may have been shortened for overview (see *Supplementary file 1b*). Tumor stage has been calculated using the Danish cancer registry and the pancreatic cancer TNM staging classification version 7. Outlier dots have been removed to safeguard patient-sensitive information.

The online version of this article includes the following figure supplement(s) for figure 2:

**Figure supplement 1.** The most frequent registry-based symptoms from the NPR prior to pancreatic cancer diagnosis.

**Figure supplement 2.** Staging information for pancreatic cancer patients.

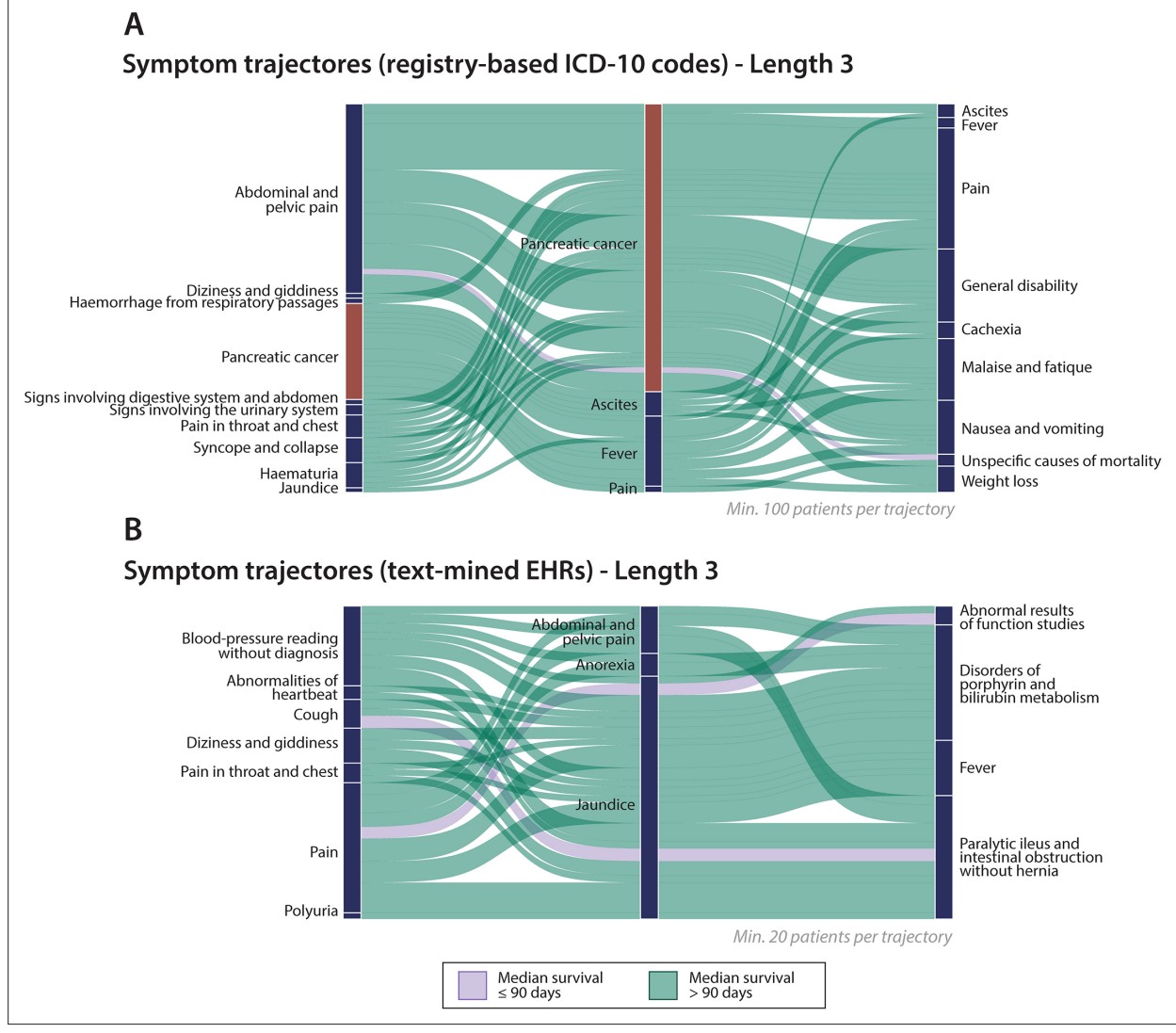

**Figure 3.** Symptom trajectories before and after pancreatic cancer diagnosis. (**A**) The registry symptom trajectories consist of significant disease pairs with a Relative Risk (RR) >1 (***Supplementary file 1d***). Each trajectory has a minimum of 100 patients. (**B**) Symptom trajectories from clinical notes consisting of significant disease pairs with Odds Ratio (OR) >1 (***Supplementary file 1e***). Each trajectory is followed by a minimum of 20 patients. The width of the trajectories indicates visually the number of patients. The purple-coloured trajectories represent patient groups with median survival ≤90 days. The green-coloured trajectories represent patient groups with median survival >90. Some symptoms names have been shortened for overview.

The online version of this article includes the following figure supplement(s) for figure 3:

**Figure supplement 1.** Registry-based trajectories with both symptoms and diseases.

**Figure supplement 2.** Survival of patients in significant trajectories.

survival patterns for the trajectories, since only a small fraction of patients are resectable at time of diagnosis (***Sgouros and Maraveyas, 2008***; ***Berkman et al., 2023***).

For the registry-based trajectories we could identify known symptoms related to pancreatic cancer such as 'Abdominal and pelvic pain', 'Syncope (fainting) and dizziness', 'Jaundice', 'Signs involving digestive system and abdomen', 'Signs involving urinary system' prior to the pancreatic cancer diagnosis. We also identified less known symptoms such as 'Haemorrhage' in respiratory pathways, 'Haematuria' (blood in urine) and 'Pain in throat and chest'. Using the NPR registry, we also analysed significant pancreatic cancer disease trajectories across all disease chapters in the ICD-10 classification system (***Figure 3—figure supplement 1***). We detected several disease associations in the cardiovascular chapter (Chapter 9) such as 'Angina pectoris' (chest pain), 'Acute myocardial

infarction', 'Chronic ischemic heart disease', and 'Hypertension'. We also found the well-known risk factor 'Type 2 Diabetes'. Other disease associations found are part of the Digestive system chapter 11 including 'Cholelithiasis' (gallstones) and 'Disorders of lipoprotein metabolism'. From both length three and four trajectories, we can observe that patients having several cardiovascular diseases prior to pancreatic cancer tend to have worse survival. We additionally looked into the temporality of the significant symptoms prior to the pancreatic cancer diagnosis (*Figure 2—figure supplement 1*). We found 'Headache' and 'Enlarged lymph nodes' to be one of the symptoms furthest away from the cancer diagnosis, 'Abnormalities of heartbeat' and 'Pain in throat and chest' being detected closer, and 'Visible peristalsis' and 'Abdominal and pelvic pain' being detected closest to cancer diagnosis.

### Pancreatic cancer trajectories from clinical notes

The symptom trajectories were generated for the pancreatic cancer patients with available clinical notes 5 years prior to the diagnosis (*Figure 3B*). All trajectories shown are significant in terms of direction with at least 20 patients following the complete path of three symptoms (*Supplementary file 1e*). In total 311 text-mined patients followed at least one trajectory which corresponds to 10% of all the pancreatic cancer cases. A patient can follow several symptom paths. Most of the trajectories begin with 'Pain'. From the Pain symptom, a large group traverses into 'Jaundice' from which most patients either end up with 'Disorders of porphyrin and bilirubin metabolism', 'Fever', and 'Intestinal obstruction'. Patient groups that follow the trajectory starting with 'Cough' leading to 'Jaundice' and ending up with 'Intestinal obstruction' have shorter median survival (median survival ≤90 days). Also, the trajectory starting with 'Pain' progressing into 'Jaundice' and ending with 'Abnormal results of function studies' shows lower survival.

Overall, the trajectories progress from generic symptoms such as 'Pain', 'Abnormalities of heartbeat', 'Dizziness' and 'Polyuria' into more known risk factors of pancreatic cancer involving 'Abdominal and pelvic pain', 'Anorexia', and 'Jaundice'. The survival distribution of the patients following the text-mined symptom trajectories (*Figure 3—figure supplement 2A*) resembles the survival distribution for the patients following the registry-based trajectories (*Figure 3—figure supplement 2B*). We also checked the survival curves of the pancreatic cancer patients with (*Figure 2—figure supplement 2A*) and without clinical notes (*Figure 2—figure supplement 2*) 5 years prior to pancreatic cancer to conclude that no obvious biases are introduced when analysing only the group of patients with clinical notes.

## Discussion

This study uncovered statistically significant disease and symptom trajectories prior to pancreatic cancer that may be further assessed as early risk factors for pancreatic cancer for screening purposes. We complemented pancreatic cancer symptoms from hospital diagnosis codes with symptoms extracted by text mining using free text in EHRs. We observed that symptoms were more abundant in the EHRs opposed to the NPR. For example, more than 2000 patients out of 3078 text-mined pancreatic cancer cases had 'Pain' according to the free text clinical notes but in the registry, it was less than 500 patients out of the 18,523 pancreatic cancer cases. On the contrary, there were some symptoms identified more abundantly in the NPR compared to the clinical notes.

From our trajectory analysis, we detected mainly symptom trajectories for patients in the longer survival group (>90 days). This could be because patients in the short survival group (≤90 days) have fewer symptoms detected and thus will not show up in the symptom trajectories. For the text-mined symptom trajectories, we could detect more symptoms prior to the cancer diagnosis than in the registry-based trajectories using the NPR, despite the text-mining cohort being much smaller.

'Jaundice' can in some cases be a symptom of a late-stage cancer and thus poor survival (*Strasberg et al., 2014*), but in other cases contribute as a clearer cancer symptom compared to the other more generic pancreatic cancer symptoms. Studies have found the diagnostic interval to be shorter for jaundice compared to for example weight loss for pancreatic cancer patients (*Walter et al., 2016*; *Gobbi et al., 2013*). It could therefore indicate a faster diagnosis but not necessarily a longer survival. We furthermore observed that disease trajectories including several cardiovascular diseases show lower survival, and it has been suggested in previous studies that cardiovascular diseases can play a role in pancreatic cancer prognosis (*Ögren et al., 2006*; *Bertero et al., 2018*; *Liu et al., 2019*).

Additionally, cardiovascular diseases have been indicated as a risk factor for certain other cancers (*Lau et al., 2021*, *Strongman et al., 2019*), and contrariwise it is also discussed whether cancer should be included in cardiovascular risk prediction tools (*Blaes and Shenoy, 2019*).

A limitation of this study is that our data is strictly hospital contacts, which may result in a set of identified symptoms and diagnoses occurring not at the earliest disease stage. By including data from general practitioners, future studies could potentially identify even earlier symptom patterns prior to pancreatic cancer. Causal patterns cannot be directly inferred from the trajectories (*Jensen et al., 2014*). Confounders could for example be medication or other risk factors that are not accounted for in the analyses. In this type of trajectory study, we can only assess the association between diseases, but actual causal relationships will need to be validated by other types of studies, such as Mendelian randomization using genetics. Furthermore, the available period for the clinical notes (2006–2016) was shorter compared to the NPR (1994–2018). A direct comparison between the two data sources can be challenging since registry terms are not coded exactly as in the free text clinical notes. One example is acute abdomen in the symptom block 'Abdominal and pelvic pain', which describes a condition with severe abdominal pain that demands immediate medical attention. A clinical note might contain text that the patient was admitted with severe stomach pain but code it as the ICD-10 code 'R10.0 - Acute abdomen' in NPR which essentially would mean the same. Our text mining procedure had a sensitivity error of 83.4% which is in the expected range for the program Tagcorpus (*Pafilis et al., 2013*; *Pafilis et al., 2015*). Also, information bias may exist in the coding procedure within NPR and it might not always be trivial to code the correct symptom or diagnosis (*Schmidt et al., 2015*; *Lynge et al., 2011*). From our study, we show that the inclusion of symptoms text-mined from clinical notes largely complemented ICD-10-coded symptoms from the patient registry.

This study showed that deep phenotypical information stored in registry data and free text EHRs can be useful for detecting temporal patterns prior to pancreatic cancer diagnosis. Furthermore, our study reveals a notable increase in the prevalence of early symptoms within EHRs suggesting the possibility of harnessing the complete potential of health data by incorporating multi-modality symptom trajectories derived from both EHRs and registry-based codes. This integrated approach provides a comprehensive representation of the symptomatology, enabling a more accurate assessment of disease trajectories and potentially facilitating early detection and intervention. Overall, our findings underscore the importance of comprehensive symptom assessment for improved prediction and patient outcomes in pancreatic cancer.

The sequence of events that leads up till the pancreatic cancer diagnosis supports that symptoms may appear in a messy and complex order. Patients start experiencing general and unspecific symptoms, which then become more specific and severe as the cancer advances. Future applications could leverage this knowledge and use AI and advanced text mining techniques to build longitudinal multi-modal trajectories for improved symptom extraction and early prediction of pancreatic cancer in the clinics. In clinical settings, this could ultimately serve as a guide for better screening procedures and improve the lives of pancreatic cancer patients.

## Methods

### Study design

For the generation of the trajectories, we used disease codes from the hierarchical International Classification of Diseases (ICD) system. ICD version 10 codes at level 3 were used for this study. The case cohort of pancreatic cancer patients was defined based on the pancreatic cancer code C25. We only included pancreatic cancer patients that had a confirmed diagnosis in the Danish Cancer Registry.

### Patient cohorts

From the Danish NPR, all hospital encounters in Denmark during the period 1994–2018 were used in the analyses, comprising 6.9 million patients. The ICD-10 Chapter 18 'Symptoms, signs and abnormal clinical and laboratory findings' was used as vocabulary for extraction of symptoms from both NPR and the clinical notes. ICD version 10 codes at level 3 were used for this study. The EHR data comprising clinical notes included all records from the Capital Region of Denmark and Region Zealand from the period of 2006–2016. The clinical notes contain information regarding a hospital encounter such as the reason for admission, findings, symptoms, operations, and treatments. Only patients with at least

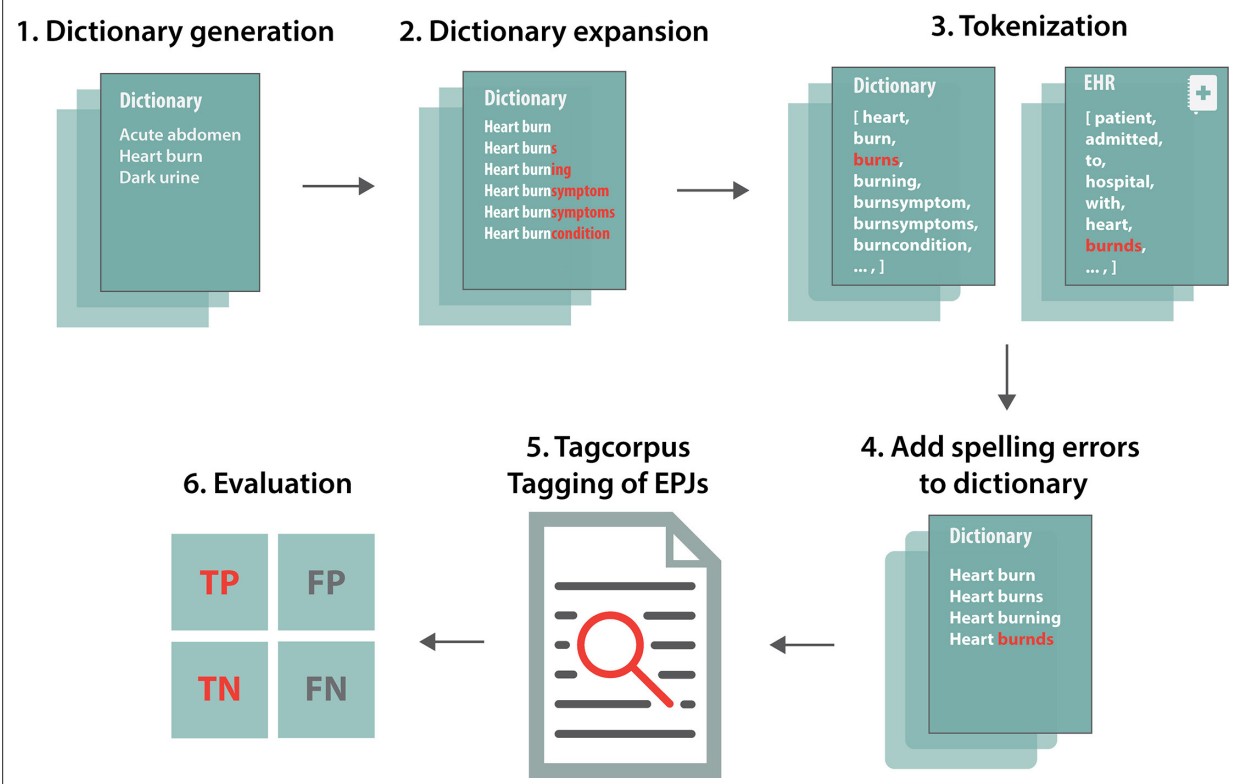

**Figure 4.** The text mining pipeline. A dictionary was generated with symptoms and expanded with word endings to capture multiple forms of the same symptom. Afterwards, the dictionary and the corpus (clinical notes) were tokenized to extract spelling errors. The spelling errors were then added to the dictionary. Finally, the program Tagcorpus (*Pafilis et al., 2013*) was used to tag the symptoms in the corpus. The text mining performance was then evaluated.

one clinical note within a five-year time interval before diagnosis were included. A control group was sampled to match sex, age, birth year and diagnosis year for a hospital contact with 10 control patients per pancreatic cancer patient. Diagnosis year was used instead of the specific date since a date limited the possible control matches. The clinical notes for the control group were additionally analysed in a five-year time interval before the hospital contact. This was done to ensure that the pancreatic cancer patients were compared with a similar age group and time-span. The registries and EHRs could be linked through a pseudo-anonymized version of the Central Personal Register (CPR) number, which all Danish citizens possess.

## Preprocessing of patient cohorts

Status codes 01 and 90 have been included in the analysis, excluding the rest of the status codes to make sure only active residents in Denmark (01) and inactive dead individuals (90) are a part of the cohort. Furthermore, diagnosis types H and M (referral and temporary diagnosis) were excluded from the data for the extraction of disease and symptom frequencies and for the construction of trajectories. This was to ensure that the analysis was based on the main medically relevant diagnoses in NPR.

## Text mining clinical notes

A dictionary was constructed by including all symptoms from the ICD-10 symptom chapter 18. Known pancreatic cancer symptoms that were not already a part of chapter 18, were added to the vocabulary by both systematic review of pancreatic cancer clinical notes and literature studies. The final symptom dictionary comprised 691 symptoms. The symptoms were initially written in their singular forms and afterwards suffixes were added to ensure different variants of a symptom such as plural forms. Other word endings like 'condition' or 'symptom' were also added since these are sometimes put behind a symptom in Danish (*Figure 4*). Afterwards, the extended dictionary and the clinical notes from the

patients and the control group were tokenized. The unique tokens from each were then compared to extract spelling errors on words longer or equal to five characters. To extract words with spelling errors, the tokens from the dictionary were fuzzy matched against the tokens from the clinical notes using the Python package fuzzysearch (*Einat, 2020*).

The metric used to measure similarities was Levenshtein's distance. In this case, a maximum value of distance between two words was set to 1 to allow for either one substitution, one deletion or one addition of a character of the word. When the fuzzy matching was complete, all outcomes were further assessed and rechecked so that wrong matches could be removed.

After the extraction of variations for the synonyms, the program Tagcorpus (*Pafilis et al., 2013*) was used to tag the corpus (the clinical notes) for the pancreatic cancer patients and the control patients. It is a fast program written in C++, which for example is able to process thousands of PubMed abstracts per second (*Pafilis and Jensen, 2016*). A post-processing step was applied that removed sentences with negations and mentioning of other persons than the patient (*Eriksson et al., 2014*). This ensures that whenever there is a negation word in a sentence the word will not be tagged. For example, not, never, and no are a part of the negation dictionary. From the other person's filter, the symptom will not be tagged if a sentence contains for example mother, brother etc. since it might relate to another person and not to the patient in question. The evaluation of the text mining was done using a confusion matrix of 200 randomly selected clinical notes. These were checked manually for incorrectly and correctly matched symptoms. Afterwards, the metrics sensitivity and specificity were calculated. The sensitivity is defined as,

$$\text{Sensitivity} = \frac{\text{TP}}{\text{TP} + \text{FN}} \tag{1}$$

where TP are the true positives and FN the false negatives. The sensitivity describes the fraction of correctly matched symptoms opposed to all symptoms. Specificity is the fraction of correctly matched words that are not symptoms opposed to all the non-symptom words in the clinical notes and is defined by,

$$\text{Specificity} = \frac{\text{TN}}{\text{TN} + \text{FP}} \tag{2}$$

## Generating pancreas cancer trajectories

We calculated significant disease trajectories for the population of pancreatic cancer patients using the methodology from *Jensen et al., 2014*. Here, the Relative Risk (RR) was calculated (*Equation 3*) for the strength of an association between two diseases in the exposed group compared to a comparison group. The exposed group was matched with the same age and sex group as the comparison group and seasonal changes were accounted for by taking samples from the comparison group disease discharge to have the same week as the disease 1 (D1) discharge in the exposed group. The count for the exposed group is denoted $C_{\text{exposed}}$ and the corresponding i'th comparison group as $C_i$ where $i \in \{1,..,N\}$ and N is the number of comparison groups.

$$\text{RR} = \frac{C_{\text{exposed}}}{\frac{1}{N}\sum_i C_i} \tag{3}$$

P-values for RR were calculated using binomial tests, where the average probability of sampling a control patient with D2 was compared with $C_{\text{exposed}}$. Afterwards, the diagnosis pairs (D1, D2) were tested for directionality with binomial tests to decide if the direction is significant for the significant disease pairs found. The patients having the direction (D1→ D2) or the other direction (D2→ D1) were counted. Patients having D1 and D2 at the same time were also counted and the total count constitutes N samples with 50% probability of having either one of the directions. The p-values were afterwards Bonferroni corrected.

To find significant text-mined symptoms, we sampled 10 control patients for every pancreatic cancer case patient. The 30,780 control patients were stratified based on sex, birth year and age at diagnosis. We ensured that the control patients were diagnosed with another disease at the same age as the cases and extracted their clinical notes five years before that diagnosis. This was done to make sure the notes for the control and the case group originated at a similar period during their

lifetime and up to a diagnosis. For the construction of the symptom trajectories the odds ratios (ORs) were calculated using a logistic regression model with all possible symptom pairs and stratified by the selection criteria for the controls.

Subsequently, each symptom pair, with OR >1 and a significant p-value <0.05, was tested for directionality and Bonferroni-corrected based on the same approach as *Jensen et al., 2014*. For the text-mined symptoms, the day of admission was used as the time registered for the symptom, since capturing the symptom as early as possible is crucial for the purpose of this study.

## Approvals

The work was approved as a registry study that does not require ethical permissions in Denmark as well as patient consent. Access to the data was approved by the Danish Data Protection Agency (ref: 514-0255/18-3000, 514-0254/18-3000, SUND-2016–50), the Danish Health Data Authority (ref: FSEID-00003724 and FSEID-00003092), and the Danish Patient Safety Authority (3-3013-1731/1/).

## Code availability

The methodology and the data analysis have been carried out using Python software (version 3.8) and R version 3.6. The code for the text mining method is available at https://github.com/larsjuhljensen/tagger, (copy archived at, *Jensen, 2023*) The key algorithm for creation of disease trajectories have been described in detail in the published studies (*Jensen et al., 2014*; *Siggaard et al., 2020*) and (*Siggaard et al., 2020*).

## Acknowledgements

We acknowledge funding from the Novo Nordisk Foundation (grant agreements NNF14CC0001 and NNF17OC0027594), the BrainDrugs (R279-2018-1145) as well as the Danish Innovation Fund (5184-00102B).

## Additional information

### Competing interests

Inna M Chen: I.M.C. reported receiving research funding and hotel/airfare reimbursement to attend global health meetings from Roche, BMS, Celgene, Genis, and an advisory relationship with Amgen and AstraZeneca. Lars Juhl Jensen: L.J.J. reports ownerships in Amgen Inc, AstraZeneca PLC and Novo Nordisk A/S. Søren Brunak: SB has ownerships in Intomics A/S, Hoba Therapeutics Aps, Novo Nordisk A/S, Lundbeck A/S, ALK A/S and managing board memberships in Proscion A/S and Intomics A/S outside the submitted work. The other authors declare that no competing interests exist.

### Funding

| Funder | Grant reference number | Author |
| --- | --- | --- |
| Novo Nordisk Fonden | NNF17OC0027594 | Jessica Xin Hjaltelin |
| Novo Nordisk Fonden | NNF14CC0001 | Sif Ingibergsdóttir Novitski |
| BrainDrugs | R279-2018-1145 | Isabella Friis Jørgensen |
| ImmunAid | 779295 | Isabella Friis Jørgensen |
| ELIXIR-Converge | 871075 | Troels Siggaard |
| RiskHunt3r | 974537 | Søren Brunak |

The funders had no role in study design, data collection and interpretation, or the decision to submit the work for publication.

### Author contributions

Jessica Xin Hjaltelin, Conceptualization, Data curation, Software, Formal analysis, Visualization, Methodology, Writing – original draft; Sif Ingibergsdóttir Novitski, Data curation, Software, Formal

analysis, Visualization, Methodology, Writing – original draft; Isabella Friis Jørgensen, Data curation, Software, Formal analysis, Supervision, Methodology, Writing – review and editing; Troels Siggaard, Data curation, Software, Investigation, Writing – review and editing; Siri Amalie Vulpius, Data curation, Software, Investigation, Methodology, Writing – review and editing; David Westergaard, Methodology; Julia Sidenius Johansen, Inna M Chen, Supervision, Investigation, Writing – review and editing; Lars Juhl Jensen, Software, Supervision, Methodology, Writing – review and editing; Søren Brunak, Conceptualization, Supervision, Methodology, Writing – review and editing

### Author ORCIDs
Jessica Xin Hjaltelin ⓘ http://orcid.org/0000-0002-7878-7830
Sif Ingibergsdóttir Novitski ⓘ http://orcid.org/0000-0003-2577-155X
Troels Siggaard ⓘ http://orcid.org/0000-0003-0235-6150
Søren Brunak ⓘ https://orcid.org/0000-0003-0316-5866

### Decision letter and Author response
Decision letter https://doi.org/10.7554/eLife.84919.sa1
Author response https://doi.org/10.7554/eLife.84919.sa2

## Additional files

### Supplementary files
• MDAR checklist
• Supplementary file 1.

### Data availability
Permission to access the person-sensitive data used for this study can be obtained through the Danish Data Protection Agency, the Danish Health Authority, and the Danish Health regions (Capital Region and Region Zealand). As the raw electronic patient records and registry information are individual-level data they are person sensitive and cannot be made publicly available but only analysed in closed, secure environments. In the paper we have only provided diagnosis and co-occurrence information when grouped to at least five patients. All patient information published here is non-person-sensitive summary level data, and can be shared per request to the authors. All processed, summarized and de-identified datasets used for plotting have been shared via Supplementary File 1.The methodology and the data analysis have been carried out using Python software (version 3.8) and R version 3.6. The code for the text mining method is available at https://github.com/larsjuhljensen/tagger, (copy archived at *Jensen, 2023*). The key algorithm for creation of disease trajectories have been described in detail in the published studies *Jensen et al., 2014* and *Siggaard et al., 2020*.

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
