## [Editor Report]

This article presents valuable findings on the symptoms and disease trajectories preceding a diagnosis of pancreatic cancer in Denmark. The evidence is convincing and the study employed appropriate and validated methodology in line with current state-of-the-art. The work will be of interest to public health researchers and clinicians working on pancreatic cancer.

---

## [Decision Letter]

**Decision letter after peer review:**

Thank you for submitting your article "Pancreatic cancer symptom trajectories from Danish registry data and free text in electronic health records" for consideration by *eLife*. Your article has been reviewed by 2 peer reviewers, and the evaluation has been overseen by a Reviewing Editor and a Senior Editor. The reviewers have opted to remain anonymous.

As is customary in *eLife*, the reviewers have discussed their critiques with one another. What follows below is the Reviewing Editor's edited compilation of the essential and ancillary points provided by reviewers in their critiques and in their interaction post-review.

Essential revisions:

This study presents valuable and solid findings on the symptoms and disease trajectories preceding a diagnosis of pancreatic cancer. Please address the major concerns of both reviewers. Revisions should include, (1) an error analysis for the text mining evaluation results with information on recall errors, and (2) a discussion on how the findings of this work could be applied in practice towards early detection and prognosis of pancreatic cancer.

*Reviewer #1 (Recommendations for the authors):*

1) It is difficult to compare results from the free-text entries and diagnosis codes, as the former are based on only a small subset of patients. If possible, the authors should examine the subset of NPR data relating to patients with free-text clinical records available. Such an analysis could help show a lot more directly what is captured by free-text entries versus diagnosis codes.

This analysis would also allow us to examine free-text notes and diagnosis codes together – if the aim is to help detect pancreatic cancer earlier, and especially if free-text notes and diagnosis code entries capture different symptoms and early-warning signs, using more of the available information could be highly advantageous.

2) "We found that for patients with short survival ({less than or equal to}90 days), symptoms could be tracked further back in the clinical notes, whereas patients with longer survival showed symptoms appearing closer to diagnosis. This could indicate that the pancreatic cancer had metastasized further in the patients with short survival.". Could cancer stage or spread at diagnosis be obtained from the NPR or clinical notes? That would allow for the testing of this hypothesis.

*Reviewer #2 (Recommendations for the authors):*

For the results in Figure 1, raw frequency of symptom occurrence was used to depict their relevance to the disease. However, it should be noted that some symptoms may be common for both pancreatic cancer patients and controls. In fact, your text mining method detected at least one symptom in the clinical notes of nearly 90% of the control group. Therefore, the authors may consider moving results in Figure 3 immediately after Figure 1 results.

Regarding Figure 3, I noticed that for each symptom, there are two color bars representing each patient group based on their median survival days. I'm curious why the purple bar is typically longer in this figure. Additionally, could you please clarify the meaning of the black dots mentioned in the figure caption, as they are not entirely self-explanatory?

Symptoms in Figure 2 are color-coded by their higher-level groups. Can this be done also for Figure 1 results, so that it makes it more obvious to tell the differences in the two data sources?

Typo in Table 1. Pancer should be cancer.

Table S2 uses specific ICD-10 codes for symptoms. For the convenience of the readers, they need to be accompanied by names. For example, what symptoms are C25, R50, and R52 in the first row?

S2. For example, it is stated that "One of the trajectory groups is formed by cardiovascular diseases (1720 patients)" but it's unclear which rows in S2 correspond to this number.

The equations in the Methods section are not displayed well in PDF.

---

## [Author Response]

Essential revision:Reviewer #1 (Recommendations for the authors):1) It is difficult to compare results from the free-text entries and diagnosis codes, as the former are based on only a small subset of patients. If possible, the authors should examine the subset of NPR data relating to patients with free-text clinical records available. Such an analysis could help show a lot more directly what is captured by free-text entries versus diagnosis codes.

We agree with the reviewer that this would allow us to compare the results more directly between the two data types. However, we performed two different types of analyses on the two data sets: (1) a population-wide study for the NPR registry data and (2) a case-control study using the text-mined data set. The purpose using two approaches was to uncover and compare potential symptoms at the trajectory-level using the entire cohorts, and not only for a limited patient-cohort.

However, to address the problem highlighted by the reviewer, we have generated symptoms trajectories for the registry-based analysis and now compare these versus the text-mining analysis, and have added the diagnosis-wide trajectories (including other chapters of the ICD-10) to the Supplementary figures.

This analysis would also allow us to examine free-text notes and diagnosis codes together – if the aim is to help detect pancreatic cancer earlier, and especially if free-text notes and diagnosis code entries capture different symptoms and early-warning signs, using more of the available information could be highly advantageous.

This could be interesting, but is out of scope for this paper. Here we would like to stress the proof-of-concept that the two data types can complement each other. The next steps would be to generate these multimodal trajectories to for example test if they are predictive of pancreatic cancer. Nonetheless, we acknowledge the significance of this perspective and have incorporated it into the Discussion section of our manuscript.

2) "We found that for patients with short survival ({less than or equal to}90 days), symptoms could be tracked further back in the clinical notes, whereas patients with longer survival showed symptoms appearing closer to diagnosis. This could indicate that the pancreatic cancer had metastasized further in the patients with short survival.". Could cancer stage or spread at diagnosis be obtained from the NPR or clinical notes? That would allow for the testing of this hypothesis.

We agree with the reviewer that this added information would be beneficial to further understand if the cancer stage plays a role. Hence, we have used the Danish cancer registry to add staging information to the patients with the text-mined symptoms (Figure 2).

Reviewer #2 (Recommendations for the authors):For the results in Figure 1, raw frequency of symptom occurrence was used to depict their relevance to the disease. However, it should be noted that some symptoms may be common for both pancreatic cancer patients and controls. In fact, your text mining method detected at least one symptom in the clinical notes of nearly 90% of the control group. Therefore, the authors may consider moving results in Figure 3 immediately after Figure 1 results.

We agree and have moved Figure 3 so that it is Figure 2 in the revised version.

Regarding Figure 3, I noticed that for each symptom, there are two color bars representing each patient group based on their median survival days. I'm curious why the purple bar is typically longer in this figure. Additionally, could you please clarify the meaning of the black dots mentioned in the figure caption, as they are not entirely self-explanatory?

(Figure 3 is now Figure 2) In this version, that does not seem to be the case any longer. We have now clarified the black dots in the figure caption.

Symptoms in Figure 2 are color-coded by their higher-level groups. Can this be done also for Figure 1 results, so that it makes it more obvious to tell the differences in the two data sources?

(Figure 2 is now Figure 3) We understand that it can be confusing, all symptoms in Figure 2 (now Figure 3) are colored in navy (since its only one chapter in ICD-10 – which has the navy color) and disease categories are colored by chapter colors. In this version of the manuscript, we chose to only showcase symptom trajectories for both data types since these are more directly comparable, as suggested by the reviewer. The chapter-colored diseases trajectories have been moved to Figure 3—figure supplement 1.

Typo in Table 1. Pancer should be cancer.

Thanks. We have corrected this typo in Table 1.

Table S2 uses specific ICD-10 codes for symptoms. For the convenience of the readers, they need to be accompanied by names. For example, what symptoms are C25, R50, and R52 in the first row?

We agree with the reviewer and the symptom codes have been converted to descriptions of diseases instead of ICD-10 codes, which can be inconvenient for the reader.

S2. For example, it is stated that "One of the trajectory groups is formed by cardiovascular diseases (1720 patients)" but it's unclear which rows in S2 correspond to this number.

In this revised version we have removed these counts from the manuscript text and they can now be found in the supplementary files.

The equations in the Methods section are not displayed well in PDF.

We thank the reviewer for noticing this and have inserted equations in a format that can be displayed well.